# Rota-Lithotripsy—A Novel Bail-Out Strategy for Calcified Coronary Lesions in Acute Coronary Syndrome. The First-in-Man Experience

**DOI:** 10.3390/jcm10091872

**Published:** 2021-04-26

**Authors:** Adrian Włodarczak, Piotr Rola, Mateusz Barycki, Jan Jakub Kulczycki, Marek Szudrowicz, Maciej Lesiak, Adrian Doroszko

**Affiliations:** 1Department of Cardiology, The Copper Health Centre (MCZ), 59-300 Lubin, Poland; wlodarczak.adrian@gmail.com (A.W.); jan.jakub.kulczycki@gmail.com (J.J.K.); marek.szudrowicz@gmail.com (M.S.); 2Department of Cardiology, Provincial Specialized Hospital in Legnica, 59-220 Legnica, Poland; mateusz.barycki@gmail.com; 31st Department of Cardiology, Poznan University of Medical Sciences, 61-491 Poznan, Poland; maciej.lesiak@skpp.edu.pl; 4Department of Internal Medicine, Hypertension and Clinical Oncology, Wroclaw Medical University, 50-556 Wroclaw, Poland; adrian.doroszko@gmail.com

**Keywords:** rota-lithotripsy, acute coronary syndrome (ACS), shockwave intravascular lithotripsy (S-IVL), rotational atherectomy (RA), calcified lesions, undilatable lesions

## Abstract

Heavy calcification remains one of the greatest challenges in the treatment of coronary artery disease (CAD), especially in subjects with an acute coronary syndrome (ACS). In the present case series study of high-risk patients with ACS, including both STEMI and NSTEMI, we performed a rota-lithotripsy—a combination of rotational atherectomy with subsequent intravascular lithotripsy—as a novel bail-out strategy to facilitate stent delivery in a tortuous calcified coronary artery.

## 1. Introduction

The etiology of acute coronary syndromes (ACS) is mainly related to the presence of “vulnerable plaques”. It is defined as a sudden rupture of a lipid plaque with subsequent release of a thrombogenic substrate, which triggers platelet activation. However, around 12% of all ACS occur in calcified culprit plaque. From an etiological point of view, they can be divided into three main subtypes: eruptive calcified nodules, superficial calcific sheets, and calcified protrusions [1]. Calcified lesions continue to be one of the most challenging interventions, where optimal angiographic results are scarcely achievable, which correlates with poorer clinical outcomes [2,3]. While the growing amount of evidence indicates improvement in the short- and long-term results of percutaneous revascularization using a new generation of drug-eluting stents (DES) in acute coronary syndrome settings [4], the efficacy of percutaneous coronary intervention (PCI) in highly calcified lesions is poorer [5]. This results partially from insufficient lesion preparation and accompanying stent failure, including poor expansion, malapposition, or fracture. This results in turn in an increased risk of acute periprocedural complications (i.e., vascular dissection or perforation) as well as in subacute or chronic in-stent thrombosis and a concomitant higher rate of in-stent restenosis. Numerous strategies aiming at the appropriate preparation of calcified plaques have been implemented [6]. They may be assigned to two main groups: balloon-dependent (non-compliant (NC), cutting/scoring, or ultrahigh-pressure balloon (OPN) catheter) and atherectomy devices (rotational, laser, and orbital) focused on removing the atherosclerotic plaque when the cutting device is directed towards the penetrating plaque. Since all the techniques have some limitations, clinical trials focused on novel methods of calcified plaque modification have been eagerly undertaken. Intravascular lithotripsy (IVL) shockwave C2 (Shockwave Medical Inc., Santa Clara, CA, USA) is a novel balloon-based coronary system for IVL, which transforms electrical energy into mechanical energy during the low-pressure balloon inflation. This plaque modifying method has been shown to be safe and effective mainly in stable CAD [7,8]. Nevertheless, the amount of data on ACS is limited [7,8,9]. So far, there is very little data on its utility in subjects undergoing primary PCI in the course of acute coronary syndrome (ACS). On the one hand, the need to make an ad hoc decision during coronary angiography, without any time left for well-balanced planning of the procedure, may unacceptably increase the periprocedural risk for complications. On the other hand, the use of more conservative strategies, particularly in cases with a large area of active ischemia, may significantly worsen both the short- and long-term outcomes. Recently, a case series regarding the successful use of shock-wave intravascular lithotripsy (S-IVL) in subjects with STEMI was published [10].

More data, including a multicenter registry, demonstrate that rotational atherectomy (RA) has similar safety and angiographic outcome sin patients with NSTE-ACS and chronic coronary syndrome, concerning both short- and long-term follow-up [11]. Some data demonstrate the successful use of RA to facilitate dilation and revascularization of heavily calcified culprit lesions in patients with STEMI [12].

In the present case series of high-risk patients with ACS, including both STEMI and NSTEMI cases, we performed rota-lithotripsy—a combination of rotational atherectomy with subsequent intravascular lithotripsy—as a novel bail-out strategy for undilatable, severely calcified coronary lesions.

## 2. Materials and Methods

The study population consisted of six carefully selected cases out of all consecutive patients with ACS who qualified for PCI at our Department of Cardiology from May 2019 to February 2021. A total of 1922 ACS patients were pre-screened during that time. In this group, we performed 32 RA and 52 S-IVL procedures. All the patients were treated in a single high-volume interventional cardiology center (conducting over 1000 PCI procedures annually) with a good experience in RA procedures (conducted in 182 ACS cases so far) and S-IVL (performed in 52 ACS cases to date). Only patients with hemodynamically significant culprit calcified undilatable lesions (at least 90% stenosis of the reference vessel diameter) were included in this study. There were no angiographic exclusion criteria regarding lesion anatomy such as the length, tortuosity, severity, or prior stent placement.

For the purpose of this paper “undilatable lesion” was defined as a lesion after unsuccessful pretreatment with a non-compliant (NC) balloon and/or due to significant stenosis, initially impracticable for S-IVL (mainly high-grade stenosis). In all involved lesions, rotational atherectomy with the successful pass of a burr through the lesion (obligatory) was performed. Finally, despite using the RA as a plaque modifier, significant (over 20% of diameter) under-expansion of the NC balloon was observed. Figure 1. provides the details on the “lesion-related” angiographic and periprocedural inclusion criteria for this registry.

Safety parameters, including coronary perforation, no-reflow, ventricular arrhythmias, and the major adverse cardiac and cerebrovascular events (MACCE) that occurred in-hospital and 30-day after primary hospitalization, were recorded. MACCE was defined as a composite endpoint including acute coronary syndrome, cerebrovascular events, major bleeding, need for repeated revascularization, or death.

## 3. Cases

### 3.1. Case 1

A 71-year-old female was admitted to the cath lab presenting STEMI of the inferior wall. A coronary angiogram revealed the acute occlusion of the dominant right coronary artery (RCA) with coexisting significant lesions of the circumflex (Cx). Percutaneous coronary intervention (PCI) was performed by the right radial approach. Arterial blood flow was restored (Figure 2(a1)) by inflation of a 2.5 mm × 15 mm non-compliant (NC) 2.5 mm × 20 mm balloon catheter; however, significant under-expansion was observed. Despite the use of an extra-support guidewire, guide extension, and additional anchor-balloon maneuver, the initial delivery attempt of the S-IVL balloon catheter was unsuccessful. Therefore, we exchanged a guidewire on a Rotawire Extra Support and performed a rotational atherectomy (RA) with successful burr (size of 1.75 mm) passage through the lesion (Figure 2(b1)). However, post-dilation with a 3.5 mm × 15 mm NC balloon Emerge (Boston Scientific, Marlborough, MA, USA) gave an unfavorable effect (significant “dog bone effect”) (Figure 2(c1)). Hence, the lesion was treated with an S-IVL 3.5 mm × 12 mm (Shockwave Medical, Fremont, CA, USA) with good expansion after 40 pulses of therapy (Figure 2(d1)). Drug-eluting stent Promus PREMIER (Boston Scientific, Marlborough, MA, USA) 3.5 mm × 38 mm (16 atm.) implantation was followed by 4.0 mm × 20 mm (20 atm.) NC balloon Emerge (Boston Scientific, Marlborough, MA, USA) post-dilatation (Figure 2(e1)). Severe disease in Cx was treated via a staged PCI procedure and, simultaneously, we confirmed the satisfying result (residual stenosis was 10%) of RCA in OCT.

### 3.2. Case 2

A 75-year-old male with hypertension, hyperlipidemia, and type 2 diabetes was admitted to the peripheral Cardiology Ward with STEMI of the anterior wall where unsuccessful PCI of the left anterior descending artery (LAD) was conducted. Afterward, the patient was passed to our Cardiac Center to perform the rescue RA of the undilatable critical lesion in the LAD. A coronary angiogram revealed a heavily calcified, subtotal long critical stenosis (lumen stenosis was 90%) of the LAD (Figure 2(a2)) with slow-flow (TIMI 1) and no other significant lesions in the Cx or RCA. We performed the PCI using a transradial 6F EBU 4.0 guide catheter. After wiring the LAD with a regular angioplasty guidewire, we exchanged it on a RotaWire ES and performed RA reaching the distal part of the LAD with a burr size of 1.5 mm (Figure 2(b2)). During the second postdilatation of the LAD by an NC balloon Trek (Abbott-Vascular, Chicago, IL, USA) 3.0 mm × 20 mm (20 atm.), an undilatable lesion in the middle part of the LAD was revealed (Figure 2(c2)). Subsequently, a 3.0 mm × 12 mm S-IVL balloon (Shockwave Medical, Santa Clara, CA, USA) was used and 7 cycles of lithotripsy were delivered (Figure 2(d2)). Two overlapping DES Resolute Onyx (Medtronic Ireland, Galway, Ireland) 3.0 mm × 38 mm (16 atm.) and 2.25 mm × 38 mm (14 atm.) were implanted with subsequent post-dilation with a 3.0 NC balloon Emerge (Boston Scientific, Marlborough, MA, USA) (20 atm.) without complication or residual stenosis and TIMI 3 flow at the end of the procedure (Figure 2(e2)).

### 3.3. Case 3

A 60-year-old male suffering from hypertension, hyperlipidemia, impaired glucose tolerance, congestive heart failure with preserved ejection fraction (HFpEF = 50%), and a CAD with a history of NSTEMI 6 years earlier (treated with PCI and DES implantation to the RCA and Cx), was transferred from a peripheral hospital. The patient complained of chest pain (class III in Canadian Cardiovascular Society Scale (CCS)) of acute-onset and the resting electrocardiogram (ECG) revealed negative T waves in the V1 to V6 leads, and the serum high-sensitive Troponin-I test was positive. The initial angiogram revealed a heavily calcified significant lesion in the left main coronary artery (LM) and subtotal proximal occlusion of the LAD (lumen stenosis over 99%) (Figure 2(a3)). Initially, the patient was referred to the local Heart Team and qualified for surgical treatment (CABG). Nevertheless, due to a lack of patient consent on the proposed surgical procedure, a PCI using a transradial 6F EBU 3.5 guide catheter was performed. After wiring a distal part of the LAD with a Fielder XT (Asahi-INTECC, Aichi, Nagoya, Japan) guidewire with the additional use of a microcatheter, the RA with burr size 1.5 mm was carried out without any complications (Figure 2(b3)). During postdilatation with the NC balloon Trek (Abbott-Vascular, Chicago, IL, USA) 3.0 mm × 20 mm we found a severely calcified lesion in the ostial part of the LAD that was undilatable (Figure 2(c3)) despite using the high-pressure inflation (22 atm.). As a result, the S-IVL 3.5 mm × 12 mm balloon (Shockwave Medical, Santa Clara, CA, USA) was then passed and 5 cycles of lithotripsy were delivered (Figure 2(d3)). Three overlapping DES Resolute Onyx (Medtronic Ireland, Galway, Ireland) were implanted from the LM to the distal the part of the LAD—4.0 mm × 15 mm (14 atm.); 3.0 mm × 34 mm (14 atm.) 2.75 mm × 34 mm (14 atm.). Finally, an additional proximal optimization technique (POT) was performed with NCB Emerge (Boston Scientific, Marlborough, MA, USA) 4.5 mm × 8 mm (18 atm.). A reasonable angiographic result (less than 10% of residual stenosis) was confirmed by the OCT imaging (Figure 2(e3)).

### 3.4. Case 4

A 66-year-old man with hypertension, hyperlipidemia, persistent atrial fibrillation, history of tuberculosis, HFpEF (EF = 50%) and with coexisting CAD that was treated in the past with PCI of the LAD + DES, was admitted to our center with NSTEMI (recurrent chest pain, elevated serum hsTnI levels). A coronary angiogram revealed mild disease in the left coronary system and a heavily calcified lesion (lumen stenosis 90%) of the mid-RCA and significant stenosis in the bifurcation of the PDA/PLA (Figure 2(a4)). Following transradial coronary intervention (7F EBU 3.5), a regular guidewire was passed into the distal part of the RCA (PDA). Using a microcatheter, we exchanged a guidewire on the Rotawire Extra Support (Boston Scientific, Marlborough, MA, USA) and performed rotational atherectomy (RA) with Rotablator burr size 1.5 mm with the successful passage of the burr through the lesion (Figure 2(b4)). After a re-exchange of guidewires, we performed a high-pressure (24 atm.) inflation of the 3.0 mm × 15 mm NC-balloon catheter Pantera Leo (Biotronik, Berlin, Germany). However, despite lesion preparation with the RA, a significant “dogbone effect” on the NC catheter in the mid part of the RCA was noted (Figure 2(c4)). Hence, we performed the S-IVL using a 3.5 mm × 12 mm catheter (Shockwave Medical, Santa Clara, CA, USA) and after 50 ultrasonic pulses, we achieved full expansion (Figure 2(d4)). In the next step, we implanted a DES 3.5 mm × 40 mm (16 atm.). No residual stenosis was presented in the control angiogram. Bifurcation was treated with two DES Orsiro (Biotronik, Berlin, Germany) 3.0 mm × 18 mm (14 atm.) and 2.5 mm × 8 mm (12 atm.) using the t-stenting technique (Figure 2(e4)).

### 3.5. Case 5

An 81-year-old female with hypertension, hyperlipidemia, paroxysmal atrial fibrillation, and chronic mild anemia of unknown origin, was transferred from a peripheral Cardiology Ward following unsuccessful PCI of the LAD (undilatable critical lesion (lumen stenosis over 90%) in the LAD) in the course of NSTEMI (Figure 2(a5)). The PCI was performed via left radial 7F EBU 3.5. Initially, we performed the RA with a successful Rota burr (1.5 mm) passage through the proximal and middle part of LAD (Figure 2(b5)). Due to under-expansion with subsequent perforation of the NC balloon Pantera LEO (Biotronik, Berlin, Germany) 3.5 mm × 15 mm (20 atm.) (Figure 2(c5)) we performed the S-IVL using a 3.5 mm × 12 mm catheter (Shockwave Medical, Santa Clara, CA, USA) and after 20 ultrasonic pulses, full expansion of catheter was obtained (Figure 2(d5)). Two overlapping DES Resolute Onyx (Medtronic, Galway, Ireland) 4.0 mm × 34 mm and 3.5 mm × 38 mm were implanted with high pressure (16 atm.). Following postdilatation with a non-compliant balloon, without significant residual stenosis and TIMI 3, flow was achieved (Figure 2(e5)).

In the postprocedural period, major bleeding occurred and the patient required transfusion of 6 units of packed red blood cells. Afterward, the patient was diagnosed with a *de novo* colorectal tumor. Following clinical stabilization, the patient was qualified for a course of neoadjuvant chemotherapy with a subsequently scheduled surgical treatment after 30 days of DAPT.

### 3.6. Case 6

A 62-year-old male with hypertension, hyperlipidemia, chronic mild anemia, hemodialyzed due to chronic kidney disease (CKD), with CAD after NSTEMI 11 years ago treated with PCI + BMS to RCA, LAD, and Cx, and NSTEMI one year ago treated with implantation of DES to RCA was admitted with the next NSTEMI. A coronary angiogram revealed chronic occlusion of the LAD and subtotal occlusion of the RCA with two separate lesions: one in the proximal part of an artery due to restenosis in DES (80% lumen stenosis) and the other subcritical (over 99% of lumen stenosis) in the distal part (Figure 2(a6)). Due to arteriovenous fistulas on both arms, we performed PCI using a transfemoral 7F AL 2.0 approach. Towards severe calcifications a successful crossing of a guidewire was achieved with Turnpike Gold135 (Vascular Solutions LLC, Minneapolis, MN, USA) microcatheter with threaded tip, providing rotational advancement when rotated clockwise.

Therefore, we exchanged a guidewire on the Rotawire-Extra-Support (Boston Scientific, Marlborough, MA, USA) and performed a successful rotational atherectomy (RA) with a Rotablator burr size of 1.75 mm (Figure 2(b6)) with subequal unsuccessful (incomplete) expansion of the NC 3.0 mm × 20 mm balloon Emerge (Boston Scientific, Marlborough, MA, USA) catheter (Figure 2(c6)). Hence, we performed the S-IVL using a 3.5 mm × 12 mm catheter (Shockwave Medical, Santa Clara, CA, USA). After the application of 40 ultrasonic pulses, full expansion was obtained (Figure 2(d6)).

Three overlapping DES-Resolute Onyx (Medtronic, Galway, Ireland) 3.5 mm × 38 mm (16 atm.), 4.0 mm × 38 mm (16 atm.), and 4.0 mm × 34 mm (16 atm.) implantation was followed by an NC 4.0 mm × 20 mm Trek (Abbott-Vascular, Chicago, IL, USA) (22 atm.) post-dilation. Finally, we obtained a satisfying angiographic result without any residual stenosis in the RCA (Figure 2(e6)).

## 4. Results

The present case-series registry included six patients (four male and two female) successfully treated with RA + s-IVL in the course of ACS with positive serum hsTnI levels (average level of 205.5 pg/mL ±141.1—[=(reference 0–14.0 pg/mL)). Table 1 provides the details of the clinical, procedural, and postprocedural characteristics. All the patients were at high risk with an average of 27.4 points, according to the Syntax Score. A rota-lithotripsy procedure, consisting of primary rotational atherectomy with subsequent intravascular lithotripsy, was performed as a bail-out strategy due to lack of full expansion of the balloon catheter after successful RA. No significant peri-procedural complications were observed. No complications including coronary perforation, no-reflow, or ventricular arrhythmia were noted. In the short-term follow-up period, including the first 30 days following the procedure, no cases of acute stent thrombosis or target lesion failure were noted. There was only one case of the in-hospital MACCE (major bleeding transfusion-requirement), no other MACCE were observed in the 30 days following the intervention. The majority of procedures were performed by a radial approach (6F or 7F).

## 5. Discussion

This is, to the best of our knowledge, the first-in-man case series study to demonstrate the efficacy and safety of the bailout combination of rotational atherectomy and intravascular lithotripsy for the treatment of undilatable lesions after successful RA in subjects with acute coronary syndromes, including both STEMI and NSTEMI cases. As far as the literature is concerned, there are only a few one-case reports [13,14,15] of combined rotational atherectomy and intravascular lithotripsy used in the treatment of chronic coronary syndromes.

Heavy calcification remains one of the greatest challenges in the treatment of coronary artery disease, especially in acute setting such as ACS. In response to this clinical problem, aggressive plaque modification before stent implantation remains an essential issue in contemporary practice [16]. One of the most efficient and widespread plaque modification technique is rotational atherectomy. However, it may lead to an increased risk of periprocedural complications [17]. Rotational atherectomy modifies hard tissue, thanks to its high speed (140,000 to 180,000 rpm) rotating diamond-encrusted burrs, which perform atheroablation via sanding/abrasion and lead to the pulverizing of calcified deposits. For that reason, RA is limited to superficial plaque modification and has no effect on more profound calcium deposits [18]. On the other hand, S-IVL is focused on deep calcium deposits and is a combination of several physical phenomena that may occur individually or simultaneously: amplitude of the pressure, stretching wave, and cavitation. All of these phemonena lead to the defragmentation of calcium nodules [19]. Therefore, both methods should not be considered as separate techniques but might be complementary to each other.

As was proven, the rate of complications can be decreased by appropriate use of the RA. Selection of smaller burr size along with an optimal procedural technique (lower speed of rotation, pecking motion technique, avoidance of decelerations >5000 rpm, short 20–30 s ablation runs, and a polishing run) can prevent numerous complications, including slow-flow and no-reflow increased platelet aggregation, periprocedural necrosis of cardiac myocytes, RA-associated stroke, and burr entrapment. Rotational atherectomy is focused on a superficial modification of the calcium burden and has no effect on the deeper calcium deposits. Therefore, the use of a small burr even with successful burr passage can be insufficient to modify thick plaques with a large deposit of calcium. Escalation of burr size usually requires an increase in diameter of guiding (which can lead to vascular access-related complications) and increases the likelihood of peri-procedural complications. An interesting approach to solve these issues may be the additional use of a novel IVL System (Shockwave, Santa Clara, CA, USA), especially when despite the use of adequate burr sizing (burr-to-artery ratio of 0.5 to 0.6) [19], the lesion is still undilatable with high-pressure inflation of the NC balloons. Such an approach can help to avoid unnecessary rescue burr size escalation or urgent CABG, which can be associated with poor outcomes.

Historically, femoral access was a preferred approach for RA. However, over the last few years, this paradigm has started to change. On the one hand, some data from numerous studies [19,20] suggest that the radial approach provides a similar safety and efficacy profile as that of the femoral access, with an ongoing decreased rate of major bleedings and access points complications. On the other hand, a recent study [21] suggests that in the setting of ACS, the majority (over 70%) of operators still prefer performing the procedure through femoral access. It may partially result from better guide support and the potential need for increasing the burr size. While most of the RA-related procedures are performed using a burr size less than 1.75 mm [22] (adequate for 7F radial catheters), the presented rota-lithotripsy technique can reduce the size of the burr to 1.5 mm (with potential subsequent S-IVL use). This could lead to an increased number of procedures followed by 6F radial access and—as a result—increase the efficacy and safety of the PCI [23]. The results from our case-series study seem to be consistent with that strategic approach.

Although we did not observe the rota-lithotripsy-related periprocedural complications, the safety concerns are not unfounded. On one hand, it has been proven that RA can increase the number of periprocedural perforations and thrombotic events [18], but on the other hand, some data suggest that S-IVL can induce ventricular arrhythmias and cause perforation of treated arteries [7,8,9,16]. Particularly sensitive is the matter of antiplatelet therapy. In a large multicenter registry focused on the RA, clinical manifestation of STEMI as well as the presence of a thrombus were a part of the exclusion criteria [11]. Nevertheless, in the literature, we can find descriptions of the successful use of RA in highly pro-thrombotic conditions [12]. In our case series, we did not use the deferred rota-lithotripsy procedure in the case of initially high thrombus burden, restored flow after balloon predilatation, or absence of ongoing ischemia. It is justifiable to postpone the procedure to extinguish the thrombotic process.

While there is no current strong evidence to support the use of rota-lithotripsy in PCI for ACS management, our early experience is favorable and provides encouraging preliminary data for future studies.

Nevertheless, there is a strong need for large multicenter, randomized, prospective studies comparing the safety and efficiency of RA and S-IVL in heavily calcified lesions (there is an ongoing rota-shock trial (ClinicalTrials.gov, accessed on 6 August 2019, Identifier: NCT04047368). What is more, future prospective trials on a greater number of subjects with long-term follow-up are required in order to confirm the safety and efficacy of the rota-lithotripsy technique. Moreover, subsequent studies are needed to define a wider range of indications (in addition to the bail-out technique) and to evaluate the long-term results using prospective intravascular imaging confirmation. Presumably, the data obtained from intravascular imaging, including the IVUS and/or OCT analyses, would provide valuable and precise information to identify the optimal subjects for primary rota-lithotripsy in the event of an elective procedure, but not in the emergency “bail-out strategy” setting presented in this study. Additionally, our data suggest that even highly-complicated plaque-modification techniques in ACS setting might be handled by radial access.

## 6. Conclusions

Based on the presented series of consecutive high-risk cases with the acute coronary syndrome (including both, STEMI and NSTEMI), we postulate that a combination of rotational atherectomy and shockwave intravascular lithotripsy followed by stenting might be a potentially feasible and viable strategy with exceptional procedural success and acceptable short-term results.

## Figures and Tables

**Figure 1 jcm-10-01872-f001:**
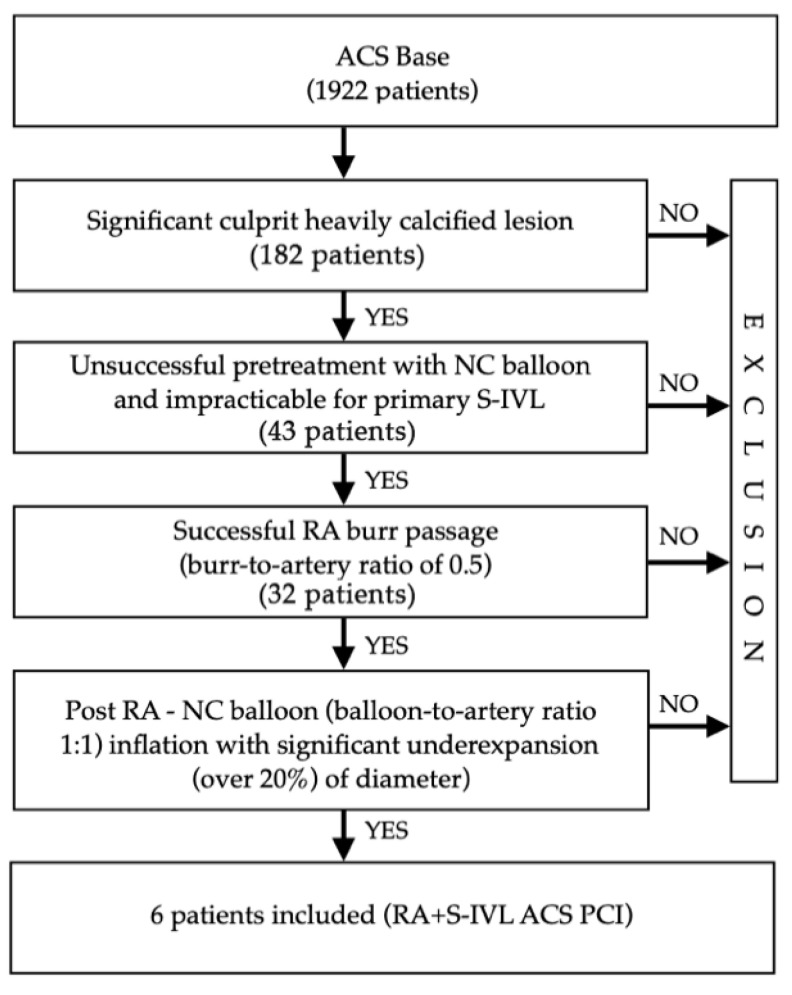
Lesion-related angiographic and procedural inclusion criteria for the study. ACS—acute coronary syndrome; NC—non-compliant; RA—rotational atherectomy; S-IVL—shockwave intravascular lithotripsy; PCI—percutaneous coronary intervention.

**Figure 2 jcm-10-01872-f002:**
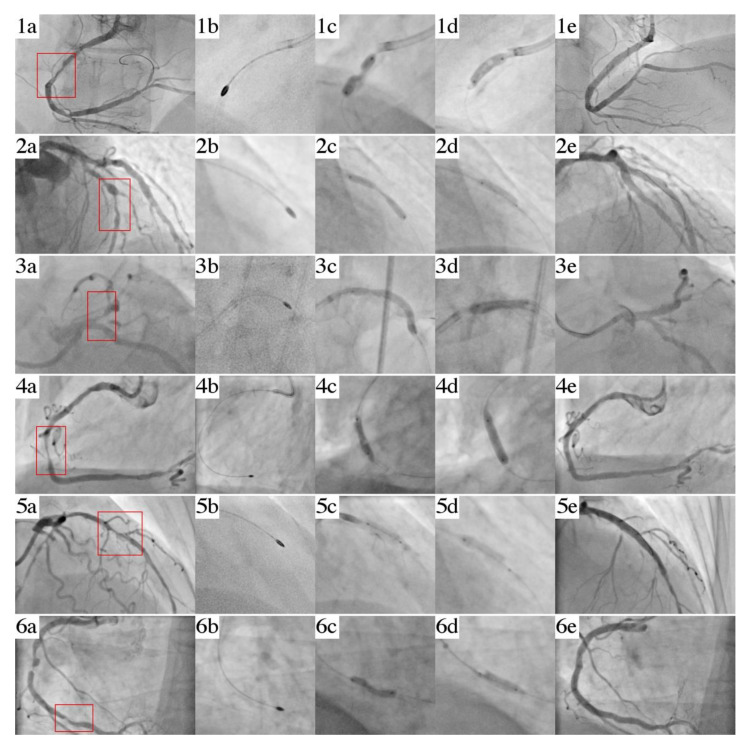
Series of cases: intervention images, Case I: (**a1**) undilatable lesion; (**b1**) rotational atherectomy; (**c1**) underexpansion of 3.5 mm × 15 mm NC balloon; (**d1**) S-IVL balloon 3.5 mm × 12 mm; (**e1**) final angiogram; Case II: (**a2**) undilatable lesion; (**b2**) rotational atherectomy; (**c2**) underexpansion of 3.0 mm × 20 mm NC balloon; (**d2**) S-IVL balloon 3.0 mm × 12 mm; (**e2**) final angiogram; Case III: (**a3**) undilatable lesion; (**b3**) rotational atherectomy; (**c3**) underexpansion of 3.0 mm × 20 mm NC balloon; (**d3**) S-IVL balloon 3.5 mm × 12 mm; (**e3**) final angiogram; Case IV: (**a4**) undilatable lesion; (**b4**) rotational atherectomy; (**c4**) underexpansion of 3.0 mm × 15 mm NC balloon; (**d4**) S-IVL balloon 3.5 mm × 12 mm; (**e4**) final angiogram; Case V: (**a5**) undilatable lesion; (**b5**) rotational atherectomy; (**c5**) perforation of 3.0 × 15mm NC balloon; (**d5**) S-IVL balloon 3.5 mm × 12 mm; (**e5**) final angiogram; CaseVI: (**a6**) undilatable lesion; (**b6**) rotational atherectomy; (**c6**) underexpansion of 3.0 mm × 20 mm NC balloon; (**d6**) S-IVL balloon 3.5 mm × 12 mm; (**e6**) final angiogram.

**Table 1 jcm-10-01872-t001:** Clinical, procedural, and postprocedural characteristics of patients.

Clinical Data	Case 1	Case 2	Case 3	Case 4	Case 5	Case 6
**Age**	71	75	60	66	81	62
**Hypertension**	No	Yes	Yes	Yes	Yes	Yes
**Type 2 Diabetes Mellitus**	No	Yes	No	No	No	No
**Hyperlipidemia**	Yes	Yes	Yes	Yes	Yes	Yes
**Atrial Fibrillation**	No	No	No	Yes	Yes	No
**Post PCI status**	No	Yes	Yes	Yes	No	Yes
**Primary Diagnosis**	STEMI	STEMI	NSTEMI	NSTEMI	NSTEMI	NSTEMI
**Treated Vessel**	RCA	LAD	LM/LAD	RCA	LAD	RCA
**Initial LVEF**	60%	44%	50%	50%	55%	35%
**Access**	7F RAD ^1^	6F RAD ^1^	6F RAD ^1^	7F RAD ^1^	7F RAD ^1^	7F FEM ^2^
**Syntax Score**	18	25	35	28	22	36.5
**Burr Size**	1.75 mm	1.5 mm	1.5 mm	1.5 mm	1.5 mm	1.75 mm
**IVL Diameter**	3.5 mm	3.0 mm	3.5 mm	3.5 mm	3.5 mm	3.5 mm
**Number of Pulses**	40	70	50	50	20	40
**DES Size/Pressure**	4.0 mm × 34 mm 20 atm.	4.0 mm × 18 mm 20 atm.	3.5 mm × 40 mm 16 atm.	3.0 mm × 26 mm 12 atm.	3.5 mm × 38 mm 16 atm.	4.0 mm × 38 mm 16 atm.
**In Hospital MACCE**	No	No	No	No	Yes	No
**30-Days MACCE**	No	No	No	No	No	No

Abbreviations: ^1^ RAD—radial; ^2^ FEM—femoral; STEMI—ST-elevation myocardial infraction; NSTEMI—no ST-elevation myocardial infraction; RCA—right coronary artery; LAD—left anterior descending; LM—left main; LVEF—left ventricular ejection fraction; IVL—intravascular lithotripsy; DES—drug eluting stent; MACCE—major adverse cardiac and cerebrovascular events.

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
