# Peer review of "Rota-Lithotripsy—A Novel Bail-Out Strategy for Calcified Coronary Lesions in Acute Coronary Syndrome. The First-in-Man Experience"

_jcm, 2021, doi:10.3390/jcm10091872_

Round 1

Reviewer 1 Report

The authors proposed a combination of rotational atherectomy and shockwave intravascular lithotripsy as a novel bail-out strategy for undilatable and heavily calcified coronary lesions in acute coronary syndrome. The paper is well presented and written. Here you have my comments:

1- I think that shockwave intravascular lithotripsy is the real and unique novel bail-out strategy in the 6 cases who are presented in the present paper. As already indicated, some previous reports already exist:

-Wong B, El-Jack S, Newcombe R, Glenie T, Armstrong G, Cicovic A, Khan A. Shockwave Intravascular Lithotripsy of Calcified
Coronary Lesions in ST-Elevation Myocardial Infarction: First-in-Man Experience. J Invasive Cardiol. 2019, 31, E73-E75

-Baudinet T, Seguy B, Cetran L, Luttoo MK, Coste P, Gerbaud E;  Bail-out therapy in ST segment elevation myocardial infarction due to calcified lesion causing stent underexpansion: Intravascular lithotripsy is in the lead. Journal of Cardiology Cases. 2021 https://doi.org/10.1016/j.jccase.2020.12.014

2- Discussion Section:

a- The vascular access is not the topic of the paper. May I suggest to not discuss this topic in the Discussion Section?

b- I would like to have some comment on the interest of intravascular imaging in this context.

c- How the authors can be sure that the choice of the burr size was adequate and sufficient in the 6 presented cases?

d- I would like to have some comment on the timing of RA and IVL in the context of STEMI and NSTEMI: interactions with thrombi? Should we defer the procedure if possible? How to choose?

3- May I suggest to improve the legend of Figure#1

Author Response

We would like to thank the Reviewer for an in-depth analysis of the manuscript and for pivotal comments provided, which have resulted in a significant improvement of this manuscript.

Comment 1   We would like to thank the Reviewer for pointing out that we can find a growing number of evidence for use of sole S-IVL in ACS – according to the Reviewer’s suggestion the two proposed references (number 9 and 10) have been added.

Comment 2a- We would like to thank the Reviewer for a suggestion regarding  the place of vascular access in the discussion section - The vascular access is not the topic of the paper. May I suggest to not discuss this topic in the Discussion Section?” We strongly believe that this topic might be valuable for readers which are the first-line practitioners in Cardiovascular Interventions, especially that vast majority of bail-out procedures in numerous case reports published in the past were conducted via femoral access.

Comment 2b  We believe that the Reviewer made a valid point asking about intravascular imaging. Hence, we added to discussion the following: “ Presumably, the data obtained from intravascular imaging, including the IVUS and/or OCT analyses, would provide a valuable and precise information for subsequent optimal subject’s qualifying for the primary Rota-lithotripsy in the event of elective procedure, but not in the emergency “bail-out strategy” setting presented in this..”

Comment 2c- How the authors can be sure that the choice of the burr size was adequate and sufficient in the 6 presented cases?

The size of the burr was matched according to the angiographic size references of treated artery in accordance with commonly accepted rule of burr sizing -  (burr-to-artery ratio of 0.5 to 0.6), [ ref : Tomey MI, Kini AS, Sharma SK. Current status of rotational atherectomy. JACC: Cardiovascular Interventions. 2014 Apr;7(4):345-53, doi: 10.1016/j.jcin.2013.12.196 ].

Comment 2d- Another valid point mentioned by the Reviewer is the influence of thrombotic aspects  in ACS on time of performing the RA + S-IVL.  Hence, we have added in the discussion section the following comment:

“Although we did not observe the Rota-lithotripsy-related periprocedural complications, the  safety concerns are not unfounded. On one hand, it has been proven that RA can increase the number of periprocedural perforations and thrombotic events [18], but on the other hand, there is some data suggesting that S-IVL can induce ventricular arrhythmias and cause perforation of treated arteries [7-9,17]. Particularly sensitive appears to be a matter antiplatelet therapy. In a large multicenter registry focused on the RA,  clinical manifestation of STEMI as well as the presence of a thrombus were a part of exclusion criteria [12]. Nevertheless, in literature we can find descriptions of successful use of RA in highly pro-thrombotic conditions [13]. In our case series, we did not use the deferred Rota-Lithotripsy procedure anyhow in the case of initially high thrombus burden, restored flow after balloon predilatation, and absence of ongoing ischemia, justifiable is postponing procedure to extinguish the thrombotic process.”

Comment 3- We believe that the Reviewer made a valid point when asking about the improving the legend of Figure 1. – it has been corrected, according to the suggestion

Reviewer 2 Report

Introduction section

I suggest to give more information about calcified coronary lesions and calcium modifying strategies in general. Additionally, more information about IVL technique and its use (as well as off-label indications) are required. It should be pointed out that heavily calcified lesions are a main issue in many patients undergoing PCI (even in chronic coronary syndrome). However, I agree that this problem can also occur in patients undergoing acute PCI for ACS.

line 27-31: please add references.

Line 31: malapposition may not only lead to stent thrombosis but also in-stent restenosis. This should be specified.

Material and methods

I think in this sections more precise information about inclusion criteria and the study follow is necessary.

Cases

In the methods sections the authors stated that they performed FFR for borderline stenosis. However, Figure 1 shows only high grade stenosis with no need for functional assessment (particularly not in ACS). This is confusing.

Results

I think, the results should be extended giving more information about baseline characteristics (sex, LV function, Troponin levels) and more detailed procedural information (reference vessel diameter, lumen stenosis, max. balloon diameter of pre and post dilatation, residual stenosis). Furthermore, more information about prior stent implantation is needed. Was RA+IVL also used for in-stent stenosis?

Line 215: “What is the noteworthy” should be deleted and transferred to the discussion section.

Discussion:

Line 234: please add reference.

The following issues may be added to the discussion section:

- use of FFR in ACS setting (if used in this case series?!?).

- Use of RA for in-stent stenosis and potential complications (in contrast to the potential use of IVL for in stent restenosis).

- Authors should also refer to the ongoing rota-shock trial (ClinicalTrials.gov Identifier: NCT04047368).

Conclusion

The conclusion should focus on the feasibility and safety during the short-term follow up. No long term results are given in this analysis.

Author Response

We would like to thank the Reviewer for an in-depth analysis of the manuscript and for pivotal comments provided, which have resulted in a significant improvement of this manuscript.

Introduction section:

As the Reviewer suggested, at the beginning of the manuscript general information about calcified coronary lesions, calcium modifying strategies and about IVL technique have added  (Line 26-38 and line 42-57). Also, part of information regarding the use of SIVL in the discussion has been added.

Line 27-31: As the Reviewer suggested, we added two references – number 2 and 3.

Line 31: We agree with the Reviewer that the stent malapposition can lead both, to the stent thrombosis and an in-stent restenosis, which is now mentioned in the manuscript, as appropriate.

Material and Methods:

As the Reviewer suggested we decided to add some more precise information about study background as well add- Scheme 1 to clarify inclusion criteria.

Cases:

We believe that the Reviewer made a valid point while asking regarding the FFR assessment in the cases described. The FFR analysis constituted a  part of the data from the two ACS registers developed in our center (ACS – RA registry including  182 cases and an ACS IVL registry – including  52 cases). The FFR assessment of borderline lesions was a part of inclusion criteria in both registries and therefore we mentioned it in the Material and Methods section. However, we fully agree with the Reviewer that in this series of cases only significant high-grade stenosis were taken into account – for that reason we decided to remove it from manuscript, since  all the presented lesions were culprit.

Results :

We would like to thank the Reviewer for pointing out that the precise information regarding clinical baseline characteristic was missing. Hence, we have added to our paper the LVEF measurement results to the Table and the summary list of the sex of patients as well as well as initial level of hs TnI. In order to avoid repetitions in manuscript missing procedural details mentioned by Reviewer were added to the individual case descriptions.

We believe that the Reviewer made a valid point when asking about the use of RA + IVL in the in-stent stenosis. All the described cases were related to lesions that had not undergone prior coronary angioplasty with use of DES or BMS or BVS. Due this fact we didn’t discuss this issue in discussion section.   

Line 215 -  We would like to thank the Reviewer for pointing that this kind of phrases shouldn’t be part of a result section – we have removed it, according to the Reviewer’s suggestion.  

Discussion:

Line 234- As the Reviewer suggested, we have added two necessary references- number 15 and 16

We would like to thank the Reviewer for pointing out that precise information about ongoing rota-shock trial was missing. Therefore, we decided to add this information to our manuscript  as follows:

“… there is a strong need for a large multicenter, randomized, prospective studies comparing the safety and efficiency of RA and S-IVL in the heavily calcified lesion (actually ongoing rota-shock trial (ClinicalTrials.gov Identifier: NCT04047368)...”

Conclusions:

We believe that the Reviewer made a valid point while indicating that this paper evaluates only short-term effects of therapy, which is now specified more precisely in the conclusions section.

Reviewer 3 Report

The present case series deal with an important topic that is the combination of Rotablator and Lithoplasty to treat heavely calcified coronary stenosis in ACS.

This study is interesting, but English language needs to be extensively improved in order to better deliver the main message.

My specific comments are the following:

Title: Consider to shorten the title, it's too long.

Introduction: A brief explanation of these two techniques separately could help non-Cardiologists to better understand the advantages of Rotalithotripsy.

Coronary lithoplasty is a relatively new technique, I suggest to add some reference about its effectiveness in ACS.

Materials and Methods: please revise your English, in some parts it's difficult to read.

Discussion: It might be interesting for the Readers to discuss the differences between lithoplasty and rotational atherectomy on their mechanism of action (ie rotational atherectomy "pulverize" calcified deposits, whereas Lithoplasty induces calcium modification).

Furthermore, the authors should also take into consideration potential side effects of these techniques.

Author Response

We would like to thank the Reviewer for an in-depth analysis of the manuscript and for pivotal comments provided, which have resulted in a significant improvement of this manuscript.

Title: As the Reviewer suggested, we have shortened the title – as follows: “Rota-lithotripsy - as a novel bail-out strategy for calcified coronary lesions in acute coronary syndrome. The First-in-Man Experience.”

Material and Methods:

As the Reviewer suggested we decided to revise this section and correct linguistic errors as well as add Scheme 1. to clarify inclusion criteria.

Introduction:

We believe that the Reviewer made a valid point while indicating that this part of paper should consist a brief description of S-IVL and atherectomy devices.  Hence, we have added the following:

“…focused on removing the atherosclerotic plaque when the cutting device is directed towards the penetrating plaque.”  and  “.. is a novel balloon-based coronary system for IVL, which transforms electrical energy into mechanical energy during low-pressure balloon inflation…. .”

As the Reviewer suggested, we have added four references regarding the effectiveness of S-IVL in ACS – number 7, 8, 9 and  10.

Discussion:

Line 215 - We would like to thank the Reviewer for pointing that this kind of phrases shouldn’t be part of a result – as the Reviewer suggested, we have removed it.  

Another valid point mentioned by the Reviewer is a suggestion to add to the discussion some information on differences in mechanism of action between  IVLS and RA.  Hence, the following sentences have been added:

“Rotational atherectomy modifies hard tissue, thanks to a high speed (140,000 to 180,000 rpm) rotating diamond encrusted burrs, which perform atheroablation via sanding/abrasion and lead to the “pulverizing” of calcified deposits. For that reason RA is limited to superficial plaque modification and has no effect on more profound calcium deposits [19]. On the other hand S-IVL is focused on deep calcium deposite, and is a combination of several physical phenomena that may occur individually or simultaneously: amplitude of the pressure, stretching wave and cavitation. All of them lead to defragmentation of calcium nodules [20]. Therefore, both methods should not be considered as a separate technique but might be rather complementary to each other.”

We believe that the Reviewer made a valid point asking the question of potential side effects of RA + S-IVL, hence we decide to raise this issue in the discussion :

“Although we did not observe the Rota-lithotripsy-related periprocedural complications, the safety concerns are not unfounded. On one hand, it has been proven that RA can increase the number of periprocedural perforations and thrombotic events [18], but on the other hand, there is some data suggesting that S-IVL can induce ventricular arrhythmias and cause perforation of treated arteries [7-9,17]. Particularly sensitive appears to be a matter antiplatelet therapy. In a large multicenter registry focused on the RA,  clinical manifestation of STEMI as well as the presence of a thrombus were a part of exclusion criteria [12]. Nevertheless, in literature we can find descriptions of successful use of RA in highly pro-thrombotic conditions [13]. In our case series, we did not use the deferred Rota-Lithotripsy procedure anyhow in the case of initially high thrombus burden, restored flow after balloon predilatation, and absence of ongoing ischemia, justifiable is postponing procedure to extinguish the thrombotic process."

Reviewer 4 Report

The authors summarized the procedures of 6 cases with ACS in whom the culprit lesion was hardly dilated by plain balloon angioplasty and was treated with rotational atherectomy and lithotripsy. Their experience may suggest the feasibility of the combination of those devices, however there are some major issues as a clinical study.

Major

It seems that they retrospectively selected the eligible cases from their pooled data. The authors should precisely describe the study flow so that the readers understand how many cases were screened and how many cases were included or excluded.

The authors should show how many ACS patients were treated with rotational atherectomy without lithotripsy and how many ACS patients were treated with lithotripsy without rotational atherectomy. In addition, they should compare the clinical outcomes among those strategies if they want to discuss on the safety and the efficacy.

In general, suboptimal dilatation is predominantly attributable to calcified lesions. The authors should assess the severity of calcification at the culprit lesions.

Intravascular images such as intravascular ultrasound or optical coherence tomography should be shown to understand the mechanisms of the undilatable lesions.

Discussion is needed for the identification of the ideal target of this strategy.

Minor

Page 1 Line 29: “Stent delivery failure, including poor expansion, malapposition,,,” In general, stent delivery failure does not include under-expansion.

Page 2 Line 55: Although the authors mentioned about FFR, the reviewer does not understand why FFR was applied to the culprit lesions of AMI.

Page 5 Line 207: Rota-Litotrypsy might be a typo.

Author Response

We would like to thank the Reviewer for an in-depth analysis of the manuscript and for pivotal comments provided, which have resulted in a significant improvement of this manuscript.

Major:

We believe that the Reviewer made a valid point when asking about the precise population background of this study. hence, we have added to the Material and Methods the following:

“The study population consist of six carefully selected cases out of all consecutive patients with ACS qualified for PCI at our Department of Cardiology from May 2019 to February 2021. A total of  1922 ACS patients were pre-screened at that time. In this group we performed 32 RA and 52 S-IVL. All the patients were treated in a single high-volume interventional cardiology center (conducting over 1000 PCI procedures  annually) with a good experience in RA procedures  (conducted in 182 ACS cases so far) and S-IVL (performed in 52 ACS cases up to date).”  As well we added Scheme 1. to clarify inclusion criteria.

Another valid point raised by the Reviewer is the concern regarding safety and the outcome of the RA and S-IVL procedure. As far as the literature is concerned, a large clinical trial comparing outcomes of the RA and S-IVL is missing. Therefore, we have mentioned this issue in the two separate parts of the discussion, as follows: 

“Although we did not observe the Rota-lithotripsy-related periprocedural complications, the  safety concerns are not unfounded. On one hand, it has been proven that RA can increase the number of periprocedural perforations and thrombotic events [18], but on the otherhand,  there is some data suggesting that S-IVL can induce ventricular arrhythmias and cause perforation of treated arteries [7-9,17]. Particularly sensitive appears to be a matter antiplatelet therapy. In a large multicenter registry focused on the RA,  clinical manifestation of STEMI as well as the presence of a thrombus were a part of exclusion criteria [12]. Nevertheless, in literature we can find descriptions of successful use of RA in highly pro-thrombotic conditions [13].”

and:

“… there is a strong need for large multicenter, randomized, prospective studies comparing the safety and efficiency of RA and S-IVL in the heavily calcified lesion (actually ongoing rota-shock trial (ClinicalTrials.gov Identifier: NCT04047368)...”

We believe that the Reviewer made a valid point while asking about intravascular imaging. In our “real-life” case study registry due to ACS setting, high complexity of PCI, and high-grade calcified stenosis of treated lesions, we did not perform an initial assessment in the  OCT/IVUS.  However, we believe, such an approach could be a valuable tool in defining an ideal target lesion feasible for Rota-lithotripsy. Hence, we have added to the discussion the following:  

“…Presumably, the data obtained  from intravascular imaging, including the IVUS and/or OCT analyses, would provide a valuable and precise information for subsequent optimal subjects’  qualifying  for the  primary Rota-lithotripsy in the event of elective procedure, but not in the emergency “bail-out strategy” setting presented in this study.”

Minor:

Line 29:  We would like to thank the Reviewer for finding this error,  of course poor expansion is not a part of stent delivery failure but rather stent failure – we corrected it.

Line 55:  We believe that the Reviewer made a valid point asking a question regarding the FFR assessment.  

The FFR analysis constituted a part of the data from two ACS registers developed in our center (ACS – RA registry including  182 cases and an ACS IVL registry – including  52 cases). The FFR assessment of borderline lesions was a part of inclusion criteria in both registries and therefore we mentioned it in the Material and Methods section. However, we fully agree with the Reviewer that in this series of cases only significant high-grade stenosis were taken into account – for that reason we decided to remove it from manuscript since all the presented lesions were culprit.

Line 207: We would like to thank the Reviewer for pointing out the typo, which has now been corrected.

Round 2

Reviewer 1 Report

The manuscript has significantly improved. Congratulations to the authors. One minor comment.

Regarding the scheme 1, I would suggest to add the numbers for each situation.

Author Response

We would like to thank the Reviewer for an in-depth analysis of the manuscript and for the pivotal comments provided, which have resulted in a significant improvement of this manuscript.

 We would like to thank the Reviewer for a suggestion regarding  Scheme 1.  as it was suggested we added information about the number of patients relating to each clinical situation 

Reviewer 3 Report

The manuscript has been improved.

Author Response

We would like to thank the Reviewer for an in-depth analysis of the manuscript and for the pivotal comments provided, which have resulted in a significant improvement of this manuscript.

Comment 1- We would like to thank the Reviewer for a suggestion regarding the language and spelling in the manuscript. We analyzed the manuscript in terms of language and style and corrected the errors